# OpenReview forum: "DataPerf: Benchmarks for Data-Centric AI Development"
_NeurIPS.cc/2023/Track/Datasets_and_Benchmarks — NeurIPS 2023 Datasets and Benchmarks Poster_

### Official Review · Reviewer_sXMf · 2023-06-22
**Excellent work!**

**Rating:** 8
**Confidence:** 4
**Clarity:** Yes

**Strengths:**

I think this work has significant contribution to the community of data-centric AI as it provides a platform to host competitions for data-centric AI, largely facilitating the research in data-centric AI. It consists of diverse tasks and well-developed tools for participation. And the involved tasks are all meaningful and match real-world applications and practice.

**Additional Feedback:**

N/A

**Correctness:**

I am not sure about this, because the authors do not describe the involved methods in details.

**Documentation:**

Yes, but the code of the five benchmarks are in separate repositories, it would be better to have a unified repository.

**Limitations:**

Yes, the authors addressed the limitations and potential negative societal impact of their work

**Opportunities For Improvement:**

While this is an excellent work, I have to point out that the current presentation is the major weakness.

It seems to me like an unorganized composition of five individual benchmarks. The motivation of these five tasks is not clear; why these five tasks? why not consider other task?

In addition, the authors do not touch much on the compared methods, which is understandable due to the page limit. But the take-away messages are very limited, as a benchmark suite consisting of diverse tasks, the author should have many insightful messages to deliver, yet what is in Section 2.1 is too general to be convincing without specific evidence and case study.

In conclusion, it could be better to 1) state the motivation for these  five benchmarks and how they collectively cover the field of data-centric AI and 2) add some insights you have gained from this benchmark suite and how these insights guide future research in data-centric AI.

**Relation To Prior Work:**

Yes

**Summary And Contributions:**

This paper presents DataPerf, a benchmark suite & competition for evaluating data-centric methods. It consists of 5 benchmarks covering a wide range of tasks and modalities. The authors also provide an open-source platform for hosting data-centric benchmarks. Overall, I think the community of data-centric AI/ML will, if not already, largely benefit from this effort.

---

> ### Author Response · Authors · 2023-08-16
> **Answer to reviewer sXMf**
>
> We thank the reviewer for their helpful feedback and recommendations.
> We address the reviewer’s concerns by section, summarizing the concern in *italics* and responding beneath.
>
> **Opportunities for Improvement**
>
> *Discuss the motivation for the set of 5 challenges*
>
> The first set of five challenges was selected from an original batch of proposals from the community. During selection, we put an emphasis on the diversity of modalities, types of tasks, and infrastructure requirements to ensure the DataPerf infrastructure could support a wide range of tasks moving forward, but we were limited by the set of proposals at the time. We hope the community can help make DataPerf a more complete offering by proposing additional challenges. We have expanded on the rationale behind selecting these five challenges in Section 2.2.
>
> *Add additional insight from the benchmark suite to guide future research in data-centric AI*
>
> During the review period we will add additional insights and lessons learned from the current benchmarks hosted by DataPerf to the Conclusion and Future Work section (6) of the paper. Additionally, we highlight that DataPerf is a community-led project to define, structure, and benchmark data-centric practices, and the DataPerf working group is intended as a forum for crowdsourcing data-centric research and methodologies, to form consensus on which facets of a still-nascent field will reward the most insights upon pursuit. The challenges comprising the first iteration of DataPerf’s benchmarking suite were selected based on working group participants’ interests and specializations, but we recognize that future work for the group will entail developing a cohesive research strategy that prioritizes developing high-impact benchmarks. We invite members of the machine learning research community to participate in shaping the future of the organization and its efforts.

---

### Official Review · Reviewer_RDrL · 2023-07-16
**Rview**

**Rating:** 6
**Confidence:** 5
**Correctness:** Yes
**Clarity:** Yes

**Strengths:**

1. Data-centric AI is a promising direction. Thus, the DataPerf project has practical values.
2. DataPerf project is managed by a strong team. A community is created to enable more researchers to get involved.
3. DataPerf is the first benchmark for data-centric AI.

**Additional Feedback:**

Data-centric AI is a promising direction, and DataPerf is an excellent effort toward this direction. However, the project seems still to be at a very early stage, and there is ample space for improvement. Currently, only five tasks are available, and they are not organized in a unified way (i.e., under different repos). Considering the existence of numerous benchmarks catering to specific tasks within the realm of data-centric AI, such as data augmentation and feature selection, the extent to which the current iteration of DataPerf can truly propel the field forward remains unclear.

**Documentation:**

The tasks are organized in separate repos without unified documentation. It is recommended to put every task under the same repo with a single documentation.

**Ethics:**

No.

**Opportunities For Improvement:**

1. Only five data-centric AI tasks are available. The selection of the tasks and the modalities seem arbitrary.
2. The code is not organized in a unified way. Each task is put in a separate repo, and they could have different APIs. Thus, the benchmark suite could not be easy to use.
3. While there is no unified benchmark for data-centric AI, there are already numerous benchmarks for many tasks related to data-centric AI. For example, there are at least 36 benchmarks (see Table 5 in [1]) for tasks like active learning [2], feature selection [3], data augmentation [4], etc. The relationship to the existing benchmarks in these individual tasks is not clearly discussed.

[1] Zha, Daochen, et al. "Data-centric artificial intelligence: A survey." arXiv preprint arXiv:2303.10158 (2023).

[2] Meduri, Venkata Vamsikrishna, et al. "A comprehensive benchmark framework for active learning methods in entity matching." Proceedings of the 2020 ACM SIGMOD International Conference on Management of Data. 2020.

[3] Bommert, Andrea, et al. "Benchmark of filter methods for feature selection in high-dimensional gene expression survival data." Briefings in Bioinformatics 23.1 (2022): bbab354.

[4] Nanni, Loris, et al. "Comparison of different image data augmentation approaches." Journal of imaging 7.12 (2021): 254.

**Relation To Prior Work:**

The relationship to the existing benchmarks in the individual tasks is not clearly discussed. See above for details.

**Summary And Contributions:**

This paper introduces DataPerf, a benchmark for data-centric AI. It introduces five data-centric AI tasks that cover different tasks in different modalities. It also initializes a community so other researchers can contribute to new benchmark tasks. The tasks are open-sourced separately in different Github repos. Some basic baselines for each task are provided for comparison.

---

> ### Author Response · Authors · 2023-08-16
> **Answer to reviewer RDrL**
>
> We thank the reviewer for their helpful feedback and recommendations.
> We address the reviewer’s concerns by section, summarizing the concern in *italics* and responding beneath.
>
>
> **Opportunities for Improvement**
>
> *Only five challenges are available and the selection of tasks and modalities seems arbitrary*
>
> The first set of five challenges was selected from an original batch of proposals from the community. During selection, we put an emphasis on the diversity of modalities, types of tasks, and infrastructure requirements to ensure the DataPerf infrastructure could support a wide range of tasks moving forward, but we were limited by the set of proposals at the time. We hope the community can help make DataPerf a more complete offering by proposing additional challenges. DataPerf will expand and evolve its offerings over time. We are currently reviewing proposals for challenges to add to the next round of submissions.
>
> We have added additional detail on the rationale for selecting our current challenges in section 2.3 of the updated manuscript.
>
> *The code is in separate repos*
>
> We have created a unified GitHub repository for all of the DataPerf challenges at <https://github.com/mlcommons/dataperf>, and we have updated our paper to link to this repository. We previously deferred this task to allow the individual organizations to develop their challenges more freely.
>
> We preserve flexibility for a diverse set of challenges by integrating them via sub-modules with specific commit hashes. This allows challenge owners to maintain control of their benchmarks and promote community visibility within their own GitHub organizations while simultaneously ensuring the challenges remain static during the competition and are archived as-is with respect to each round of challenges.
>
> *Expand the related work and discuss the relation to previous data-centric benchmarks*
>
> Thank you for highlighting these works. With the additional allotted page, we have expanded the related work in section 3 and discussed how our benchmarks and platform compare to the references pointed out here. We seek to expand DataPerf to incorporate existing benchmarks which can take advantage of the long-term support and shared infrastructure of MLCommons, and we hope our leaderboards will aid in centralizing state-of-the-art and reference solutions.
>
> **Summary of our edits to this response (August 23):** We have updated our response with a link to the unified github repository requested by the reviewer, and to reflect the current section numbering of our paper after a reorganization.

---

> > ### Comment · Reviewer_RDrL · 2023-08-24
> > **Thank you for the response**
> >
> > The majority of my concerns have been addressed. I will increase my score accordingly. However, the unified GitHub repository lacks a consistent API design. This could pose scalability challenges as more data-centric tasks emerge in the future. Therefore, I encourage the authors to dedicate more efforts toward reorganizing the DataPerf benchmark in a more unified manner.

---

> > > ### Author Response · Authors · 2023-08-27
> > > **Thank you to Reviewer RDrL**
> > >
> > > We sincerely appreciate the reviewer’s feedback and the raised score. The MLCommons organization has allocated engineering resources to make additional improvements to DataPerf and in conjunction with the Working Group, we will seek to offer consistent APIs in future challenges and benchmarks, particularly for multiple challenges in a shared domain such as coreset selection.

---

### Official Review · Reviewer_qPz4 · 2023-07-19
**Review DataPerf**

**Rating:** 6
**Confidence:** 5
**Correctness:** I found no issues about correctness.

**Strengths:**

- Data-centric Deep Learning is a very important topic and thus it is very interesting to see more research in this are. The proposed benchmark could provide such opportunities.
- The paper provides a wide variety of tasks and highlevel overviews about the tasks
- The paper is easy to read and provides a good motivation for the problems.

**Additional Feedback:**

I see the great effort especially technical effort to create this software suite and the proposed tasks. However, as reviewer I see my task in judging the provided paper and this is in the current form not acceptable to a high tier conference as Neurips. I believe this paper is suited for this track and based on the rich technical suite in the background it could also be accepted. In the current state, a lot of content is missing and included in a very high level overview.

**Clarity:**

All information that is included seems to be correct. However, a lot of information is not part of the proposed paper and rather part of external websites. As stated above a copy of this information should be included, ideally in a summarized form to allow a fair evaluation of the clarity for the reviewers and a future readers.

**Documentation:**

The documentation seems to good but the content is too much to verify everything. I checked some documentations in detail which seemed to be fine.

**Limitations:**

The authors described the limitations of current approaches but not of their work.

**Opportunities For Improvement:**

- Speech selection task: Why are so few baselines provided and even the online submission of cleanlab is not present? Why is the description of the baseline method not in the paper?
- Vision Selection task: The details about the baseline methods are not given except for the name and a weblink. A verification is difficult due to the fact that the complete website has to be checked and the main details are not provided.
- Debugging for Vision Task: The details and motivation for the embedding space are missing. I would be interesting to understand the selected verification method. As all other parameters except from the priority, it is important to verify that the verification method is technically sound. As the test set is concealed I see no reason for realising and documenting this part. Datascope is cited, it would be great to include atleast some sentences about the method if this is your baseline.
- Adversial Nibbler: I really like the idea of collecting this data and analyzing it, however, in the paper only the idea is proposed. This is not enough for publication at Neurips. This is a publication for the future when the data is available or small test runs should have been included for this work. At this point, I can only state it is a nice idea but any issues or kind of data which will be collected is in the unforseeable future.
- Details about the datasets are only available at the websites, they are not included in the paper and thus might be subject to change. I'm aware including these information is alot of work but it would be very beneficial to include this information to have a fixed reference if this paper will be accepted.
- I heard from multiple reseachers that questions have not been answered which were forwared to the DataPerf Team, thus I tried my self with a fake account to propose a topic to DataPerf but have not received an answer until the review submission deadline. This questions the claim of user-driven benchmark.
- Related Work: It is unclear what the difference to previous benchmarks is, Some benchmarks like DCBench seem to fit very well into the concept of DataPerf. It is questionable if just another benchmark is a good motivation for a new benchmark suite. If others already proposed benchmark why is the opportunity of unifying them in the new suite not discussed. Futhermore, benchmarks or datasets about visual computer human alignment such as CIFAR10H Peterson et al: Human uncertainty makes classification more robust 2019, DCIC Schmarje et al: Is one annotation enough? A data-centric image classification benchmark for noisy and ambiguous label estimation 2023 are not discussed but seem to be a missing part in the related work of Data Perf.
- Few comparisons: The paper provides few baselines I understand that a new benchmark or suite of benchmarks can not provide the best methods, however at least 5 methods per benchmark would be desirable. These baselines allow a better comparison of newer models and may help to indentify overcomplicated or oversimplified problems.

**Relation To Prior Work:**

Related work is discussed briefly but differences or similarities to previous work is not discussed.

**Summary And Contributions:**

The authors provide a software suite for data-centric tasks. Futhermore they propose 5 initital tasks in their suite. They motivate and highlight the importance of data-centric deep learning. The contributions are this software suite and the proposed five tasks.

---

> ### Author Response · Authors · 2023-08-16
> **Answer to reviewer qPz4**
>
> We thank the reviewer for their helpful feedback and recommendations. We address the reviewer’s concerns by section, summarizing the concern in *italics* and responding beneath.
>
> **Opportunities for Improvement**
>
> *Speech Task: Why are there so few baselines provided and why is the description of the baseline not in the paper?*
>
> In section 2.2.1 we have expanded the existing speech task baseline description, and added a second baseline from Cleanlab. Our intent is for submissions in the current round of a DataPerf challenge to serve as baselines in future rounds.
>
> *Selection for Vision Task: sufficient detail is not provided*
>
> We have updated the selection for vision repository with code for a baseline method, and two additional baselines will be open-sourced during the review period. We have also expanded the description of each baseline in Section 2.2.2.
>
> *Debugging for Vision Task: Add details on the embedding space, the selected verification method and the baselines.*
>
> In Section 2.2.3 we have expanded on the decision for using embeddings to direct focus to domain-agnostic feature selection methods. We describe our  incremental verification approach, and added a note on DataScope’s estimation of Shapley importance values.
>
> *Adversarial Nibbler: Good idea, but there is no data currently available, therefore it should not be included.*
>
> The Adversarial Nibbler challenge is currently ongoing (hence its inclusion in this paper). Though data analysis is not complete, it forms an important part of the DataPerf suite of challenges, as it demonstrates the flexibility of the platform and the approach to encompass both generative models and data collection efforts. We expect takeaways from this challenge to be published in the proceedings of the AACL Art of Safety workshop (where Adversarial Nibbler is a challenge track); its inclusion here serves as a proof of concept for the capabilities of the DataPerf platform. We have edited section 2.2.5 to highlight this particular methodological contribution, moreso than the dataset itself.
>
> *Details about the datasets are only available on the websites, not in the paper, and are therefore subject to change. Please include this information to have a fixed reference for review.*
>
> As DataPerf is intended to be an evolving benchmark we were concerned that some technical details would change in the future and we would be unable to update them in archived proceedings, therefore we linked to webpages that can be kept up to date to avoid confusion.
> In order to create a fixed reference, we will add all key information from these websites to the appendix with the access date (and a note that the information could be out of date in relation to the future iterations of each benchmark) prior to the end of the review response period.
>
> *The DataPerf working group did not respond to a topic proposal. This questions the claim of a user-driven benchmark.*
>
> We apologize for not responding to the proposal. In order to avoid this occurring in the future, as well as to accommodate the large number of proposals we have received since forming DataPerf, we have changed our process for proposing new challenges. DataPerf has added documentation to https://www.dataperf.org/ with a link in the paper (in section 6) to clarify and formalize the process, which involves joining the working group, sharing a proposal doc for comments, and presenting the proposal at a working group meeting. This process is similar to other MLCommons benchmarks such as MLPerf.
>
> *Related Work: Discuss the relationship to previous data-centric benchmarks, including some missing citations. Discuss the opportunity of unifying existing data-centric benchmarks*
>
> Thank you for highlighting these works, we have added them to related work (Sec 4) and have discussed their similarities and differences to the DataPerf Challenges. We believe DataPerf can serve as a unified benchmark suite for the majority of data-centric use cases, therefore we welcome proposals from the creators of existing data-centric benchmarks. Existing benchmarks that join DataPerf can benefit from shared infrastructure and will be maintained and adapted over time by the working group and the MLCommons organization.
>
> *Include five baseline methods per challenge*
>
> In our initial round of challenges we intended our baselines to provide (1) instructive starting points for participants to make contributions easy, (2) a lower bound to give other submissions context, and (3) enough rigor to outperform random submissions. Since DataPerf is built around live, public leaderboards, previous participant submissions become baselines for future rounds as well as potential inclusions as baselines in future challenges DataPerf will host. In our related work section, we have included these as our objectives in comparison to existing state-of-the-art approaches in data-centric benchmarks. We have included additional detail on the baseline solutions to each challenge in Section 2.

---

> > ### Comment · Reviewer_qPz4 · 2023-08-23
> > **Thx for the improvements**
> >
> > I acknowledge the given explanations and improvements in the document (including colored highlights of changes).
> > I see that most of my concerns are addressed while not completly resolved. The issues like more baselines and connection to other benchmarks are given and extended but not of the quality I would expect. Adversial Nibbler should be in my opinion not be part of this paper at the current stage as it is still under development.
> >
> > Nevertheless, I can see the given arguments. I reevaluated the paper not based on beeing a collection of benchmarks but a tool set for other benchmarks. In this point, the paper achieves the goal mainly but  is then cluttered with five rather long examples. I have to question what is the goal of this paper? Do you want to be a " a community-led benchmark suite for evaluating ML datasets and data-centric algorithm" as stated in the paper or do you want to introduce multiple data-centric benchmarks. If it is the first, presenting multiple benchmarks is mixing it quite a lot which results in the issues like soon outdated documentation (if all details are given now) and outdated baselines (if the current results are used). If it is the second, all 5 benchmarks are not described enough and evaluated in detail. Maybe one paper per benchmark would be necessary. It seems to me you try to achieve too many elements in the same paper. Even more, how do you want to include more benchmarks. Do you plan to release additional papers with more collection of benchmarks or is this your main white paper? Please explain how do you justify this mix of goals which can in my opinion not be fully covered in just one paper.
> >
> > I increased the score to reflect the already provided improvements and explanations.

---

> > > ### Author Response · Authors · 2023-08-24
> > > **Response to Reviewer qPz4 feedback**
> > >
> > > We thank the reviewer for their additional feedback and questions, and share our responses to each summarized point below:
> > >
> > > *improve the quality of the related work*
> > >
> > > We have made a second revision to the related work (Sec 3) to emphasize the unique contributions of DataPerf and further delineate it against existing efforts in data-centric research and machine learning benchmarking. We also ask if there are specific remaining concerns to be addressed here.
> > >
> > > *Adversarial Nibbler is still under development*
> > >
> > > We wish to clarify that the challenge is not under development, as it is an active running challenge with 227 public submissions as of August 23. We included this challenge as an example for other generative AI tools to assess the capabilities and safety of their systems using DataPerf, and as a first critical step in addressing the many gaps in evaluation methods for generative AI.
> > >
> > > *what is the goal of this paper?*
> > >
> > > Our goal is to establish community-led benchmarks (the first item). We have reorganized the paper to emphasize that the DataPerf platform is our primary contribution, by moving it to Section 2.2, and describing the challenges afterwards in Sections 2.3.1-2.3.5. To seed the DataPerf effort, we needed these five challenges to exercise the hosting requirements for the platform across very different modalities, and as demonstration efforts to be shepherded through the Working Group’s process (as all five challenges are community contributions each from a different organization). The Working Group performed integration cycles with each challenge, to ensure DataPerf supports a wide swathe of the machine learning community’s needs.
> > >
> > > In our latest revision, for each benchmark we have also added detail on what aspects of the DataPerf platform it exercises, to help future challenge authors understand the capabilities provided by DataPerf within their domain (Sec 2.3.1-2.3.5).
> > >
> > > *this paper will suffer from outdated documentation and outdated baselines*
> > >
> > > We clarify that the documentation for our benchmarks will not become outdated, as these benchmarks are fixed, static targets, and researchers are encouraged to use these benchmarks to compare current and advance future data-centric methods. The baselines we introduce are initial solutions to the challenges and we expect them to be superseded, as is common in the benchmarking community.
> > >
> > > In our new revision, in Sec. 2.3 we clarify our distinction between challenges, benchmarks, and baselines as they relate to the goals of our paper. Challenges are time-windowed events which provide calls for participation and opportunity for recognition, while benchmarks are the long-term leave-behinds for each challenge, accompanied by a permanent leaderboard which will continue to accept submissions. We feel this long-term infrastructure support is a useful contribution in comparison to many benchmarking efforts.
> > >
> > > *the paper is cluttered with five long examples*
> > >
> > > Our aim is to provide audiences from each community in vision, speech, sample importance estimation, data markets, and generative AI with a clear, representative example of hosting a challenge on DataPerf. We have tried to convey enough technical detail to persuade stakeholders in each of these domains to contribute similar tasks with their existing datasets to the DataPerf suite. We also hope the diverse challenges convey the flexibility of our platform to expand to other domains. We have moved some details and documentation for each benchmark to the Appendix, and additionally expanded our documentation in the Appendix so that it can serve as a fixed and centralized reference for the current benchmarks in the suite.
> > >
> > > *all 5 benchmarks are not described enough and evaluated in detail*
> > >
> > > We have expanded our Appendix in the supplementary material with additional documentation on each benchmark, and will continue to expand this documentation through the review period.
> > >
> > > *is this your main white paper?*
> > >
> > > Yes this paper is intended to serve as the primary source for the platform design and methodology of the working group. Additional benchmarks and challenges will be introduced on the platform without requiring a concurrent or existing publication, though the authors of current and future challenges may wish to publish follow-on work to analyze received submissions relative to a specific task if merited by novelty.

---

> > > > ### Comment · Reviewer_qPz4 · 2023-08-25
> > > > **Reply**
> > > >
> > > > First of all, thank you for your speedy reply. I understand now better what the goal of your work is and believe your work improved again. Thus, I will raise my score again and hope that your work will be accepted.
> > > >
> > > > For me, still some issues are remaining which maybe subjective in nature and they would require a more drastic rewriting as before.
> > > > - Adverserial Nibbler is an ongoing project and I think it should not be part of this paper. There are no gained insights except that your platform can support such challenges. I understand your reasoning but just disagree on the design choice as it weakens in my opinion the paper.
> > > > - The evaluation of the five/four tasks are short which is intended as you describe but I would prefer if it gave me some full evaluation including a diverse set of baselines. This would require more likely one paper per task.
> > > > - The connection to other benchmarks is presented but if the goal is to be a universal platform more effort should be put into integrating all other (smaller) benchmarks. This is very costly demand but would elevate this paper from acceptable to outstanding.

---

> > > > > ### Author Response · Authors · 2023-08-27
> > > > > **Thank you to Reviewer qPz4**
> > > > >
> > > > > We thank the reviewer for their valuable feedback on the additional improvements to be made in DataPerf, and for raising the score. With the engineering support the MLCommons organization has allocated for new and continuing challenges hosted on DataPerf, the Working Group will extend the number of available baselines (with more comprehensive evaluations provided in our documentation) and we will integrate additional benchmarks from the cited references.

---

### Official Review · Reviewer_KK87 · 2023-07-20
**Great project, but would have been interested in knowing more about how it was arrived at.**

**Rating:** 8
**Confidence:** 4
**Correctness:** Yes.

**Strengths:**

* The paper describes a platform to test data-centric AI techniques. This makes a lot of sense, and has been discussed a lot, so it's great to see realisations coming out.
* The paper is well written, clear and concise
* It presents a variety of use-cases, and for each a variety of Baselines. The use-cases and rationale behind them are well explained.
* The authors accept submissions of new data sets for the platform, which addresses most of my "opportunities for improvement" comments.

**Additional Feedback:**

Just a question, sometimes people intend "model" to refer to the code+parameter values, and sometimes to just the code. Did anyone come up with a word that is unambiguous?

**Clarity:**

The paper is generally well written and clear, however it would be good to check consistency between the terms in the paper and figures of the various "training" elements: "Selected Training IDs", "Allowed Training IDs", "Training Set", "Training Data", "New Train Set", etc. Sometimes one has to think which exact set is being referred to.

**Documentation:**

Very good.

**Ethics:**

I saw no issues.

**Limitations:**

Yes.

**Opportunities For Improvement:**

some choices in limiting the scope of the platform (and paper) were made, and although I don't expect the scope to be expanded, I expect an explanation of what the reasoning was behind making the choices you have made. In particular:

1) Data-centric AI can be seen as a "holistic" approach, including data capture, data pre-processing (the "data-centric operations" in the paper), and optimisation of the code part (model in the paper) to make the most of the "data-centric" element. A typical use-case may be to bring the overall costs of running the model down, without performance degradation, by appropriately normalising the input data. This paper chooses to focus most on the "data-centric operations", which is fine, but it would be great to understand the applicability of this benchmark to typical workflows of data-centric projects, where the entirety of the pipeline, from capture to model is under the control of the developer.
2) Data-centric AI allows you to get more out of less data that is of higher quality and better defined. Typical examples may be in medical diagnostics, where there may be a only a small sample of data (e.g. rare illnesses), or industrial processes, where the data comes from a single manufacturer, and labelling is expensive. I felt that the Use-cases that are cited are more common of "model-centric" ai, i.e. consumer applications where lots of data of varying quality is used. It's a pity that none of the use-cases use some of more advanced data-centric technique, such as raw data with physical synthesis models.
3) If I understand correctly: in most of the benchmarks you get a pool of data, from which you can pre-select some "Selected training IDs", that you then feed as the input data "training data" in figure 1, and therefore to "data-centric operations" that can generate new training sets at will. Is this right? Maybe consider ensuring coherence between Fig 1, Fig 2 and the text.

**Relation To Prior Work:**

Yes, very clearly.

**Summary And Contributions:**

I have seen several proposals for  benchmarking of data-centric AI, these are absolutely necessary to improve research.
This paper presents the results, however, in my opinion, as data-centric AI is still in its infancy, it is lacking several elements:

1) A clear explanation on the reason for the choices that were made in making the platform, and features that were included/not included, as well as the choices that were made between generality and usability.
2) I feel that the Use-cases are more typical of standard "model-centric" AI ones, i.e. large quantities of non-qualified data available for consumer applications, rather than the more typically data-centric AI ones e.g. industrial/medical/technical applications with little data that is very well qualified.
3) Baseline methods are described, but it would have been interesting to at least one of the examples explained in more detailed, so that the reader can get a better feeling for how the system performs in practice, as well as how the limits of the platform impact the overall performance.

---

> ### Author Response · Authors · 2023-08-16
> **Answer to reviewer KK87**
>
> We thank the reviewer for their helpful feedback and recommendations.
> We address the reviewer’s concerns by section, summarizing the concern in *italics* and responding beneath.
>
>
> **Summary and Contributions**
>
> *1. A clear explanation of the design choices that were made*
>
> Before the review response period ends we will add additional context on the design choices of the platform in section 3. Specifically, we will discuss how the platform was designed to balance flexibility and support while leveraging existing Dynabench infrastructure where possible.
>
>
> *2. The Use-cases are more typical of standard "model-centric" AI ones (large datasets).*
>
> The first round of challenges is reflective of the proposals we were able to solicit from the data-centric community, and we have expanded briefly on why these use cases were chosen in Sec. 2.2. We aim to include more use-cases in future challenges.
>
>
> *3. Additional detail on baseline methods.*
>
> We have added additional detail on each of the baseline methods in Section 2. Additionally, the baseline solutions and results are open source and serve as the starting point for prospective participants in the challenge.
>
>
> **Opportunities For Improvement**
>
> *1. DataPerf focuses on “data-centric operations” but does not capture the end-to-end pipeline, from data capture to model.*
>
> Dataperf focuses on data-centric operations as they were underrepresented in past ML benchmarks, despite being critical in modern ML pipelines. Competitions that target the entire pipeline are valuable, but we currently aim to focus efforts on individual stages which increases the direct comparability of solutions on a leaderboard.
> While data capture may be difficult to adequately represent in a benchmark, DataPerf welcomes proposals on data capture challenges. We note Adversarial Nibbler may be a step towards including data capture in DataPerf, as the participant generates new adversarial prompts in order to identify blindspots in image generation safety filters.
>
>
> *2. Dataperf focuses on "model-centric" data use cases where there is lots of data. This omits use cases where the goal is to get more out of smaller, high-quality datasets. These use cases are common in the medical domain.*
>
> The current selection of challenges reflects the first round of proposals from the community. We aim to expand the scope of DataPerf to serve additional domains (including low-resource, healthcare, and manufacturing settings), and to target these currently omitted use-cases. Our working group solicits experts with an established dataset and use case to propose challenges. We also note that the vision selection challenge involves selecting from a large pool of unlabeled images given a small number of examples, which has partial overlap with the medical use case mentioned.
>
>
> *3. Ensuring coherence between Fig 1, Fig 2, and the text. Specifically in regards to “Selected training IDs” vs. “new training data”*
>
> Figure 2 only illustrates the system design of the speech selection challenge and is not representative of the other four challenges. We use “Selected training IDs” in Figure 2 to match the code of the speech challenge, which passes IDs instead of data samples for efficiency.
> Figure 1 is meant to illustrate the overall data-centric pipeline and highlight the data-centric operations we aim to benchmark. We have updated the “Representation Selection” text in Figure 1 to “Data Selection” to be consistent with the naming convention elsewhere.
>
>
> **Clarity**
>
> *Check consistency between the terms in the paper*
>
> We will ensure the terminology of the paper is consistent and will add a glossary to the appendix to resolve any ambiguity.

---

### Author Response · Authors · 2023-08-30
**Summary of revisions**

We sincerely thank all reviewers for their insightful feedback and helpful responses. Below we summarize our changes and improvements based on reviewer suggestions (with key revisions also highlighted in blue text in the PDF).

* Additional context on the DataPerf hosting platform design choices (Section 2.2) and rationale for why these challenges were selected as our initial benchmarks by the DataPerf Working Group (Section 2.3)
* Expanded technical details on our baseline implementations for each benchmark (2.3.1-2.3.5).
* Expanded related works section (Section 3)
* Unified the code for each baseline implementation under a single GitHub repository (https://github.com/mlcommons/dataperf).
* Incorporated all key documentation from each benchmark’s GitHub readme into the Appendix as a centralized reference
* Added a link to the form for proposing new challenges on DataPerf.org (Section 5).
* Clarified our terminology for challenge, benchmark, and baseline in Section 2.3 and training set selection in Appendix A1.

---

### Decision · Program_Chairs · 2023-09-22

**Decision:**

Accept (Poster)

**Comment:**

This paper proposed a new data-centric community-based benchmark suite, DataPerf. All four reviewers gave beyond acceptance scores while some reviewers pointed out the concerns on data selection motivation. The authors sufficiently addressed the motivation of how to select tasks considering both community proposals and modality balance.

The AC raised a concern of no data contribution. While data is an important factor for decision, the D&B track covers the contributions of well-designed benchmark as well as dataset itself. Considering the reviewers' score and the author response, DataPerf can contribute to the ML community significantly, which all reviewers agree. Therefore, the SAC recommends acceptance.